# Study on Water Vertical Infiltration Characteristics and Water Content Simulation of Sandstone Overlying Loess

Xiaoyu Dong [1], Fucang Qin [2,*], Long Li [1], Zhenqi Yang [3], Yan Li [1] and Yihan Wu [1]

1 College of Desert Control Science and Engineering, Inner Mongolia Agricultural University, Hohhot 010000, China
2 College of Forestry, Inner Mongolia Agricultural University, Hohhot 010019, China
3 Institute of Water Resources for Pastoral Area, Ministry of Water Resources of China, Hohhot 010020, China
* Correspondence: qinfc@126.com

**Abstract:** Research on the infiltration characteristics of surface water is fundamental for understanding the entire hydrological process. Therefore, studying the water infiltration process of sandstone slopes overlaid with loess and predicting soil moisture content are of great importance for investigating hydrological processes and controlling soil erosion in the hilly and gully areas of the Loess Plateau in China. This study mainly focuses on the simulation of the vertical water infiltration characteristics and water movement patterns of four kinds of sandstone (feldspathic and argillaceous sandstone) structures covered with thin layers of loess. In the one-dimensional vertical infiltration experiment and Hydrus-1D model simulation, the interlayer transition planes of loess–feldspathic and loess–feldspathic–argillaceous sandstones were found to present two conditions: fine soil covering coarse soil and coarse soil covering fine soil. Therefore, water infiltration reduced permeability. The existence of a transition layer between loess and feldspathic sandstone decreased the water infiltration rate and infiltration amount and decelerated the speed of the wetting front, thereby further affecting the ability of water infiltration. By using the Hydrus-1D model, 15 sets of soil hydraulic parameters, including $\theta x$ (0.028–0.05795 cm$^3$/cm$^3$), $\theta s$ (0.2306–0.4786 cm$^3$/cm$^3$), $\alpha$ (0.01899–0.06071 cm$^{-1}$), $n$ (1.438–6.408), and Ks (1.96·10$^{-4}$–0.0576 cm/s) were inverted and optimized for each 20 cm soil layer (total of 60 cm). The Van Genuchten model constructed using these parameters demonstrated high accuracy in the simulation of water content in the vertical infiltration process of sandstone covered by loess with the coefficient of determination R$^2$ > 0.849 and relative error RE < 5.311.

**Keywords:** surface water infiltration; Hydrus-1D; sandstone; loess hilly and gully region in China

## 1. Introduction

Surface water infiltration refers to direct or indirect water transport through the surface to deep soil layers [1]. During this process, large amounts of water are stored in the soil and rock pores. Therefore, surface water infiltration is an important link in the mutual transformations of natural rainfall, surface water, soil layer water, and groundwater and is an important basis for the study of solute transport and soil erosion [2–5].

Differences in soil and rock textures lead to discrepancies in the process and characteristics of water infiltration [6]. These discrepancies are caused by the joint action of physical properties, such as porosity and bulk density, and the mechanical composition of soil and rocks with different textures [7]. Numerous scholars have analyzed and simulated the trend of water infiltration under various geological conditions, including water transportation in sandy soil [8], loam soil [9], silty rocks [10], and sandy rocks [11]. However, most previous studies investigated water transportation under the uniform conditions of the same soil texture, and studies on water infiltration under different soil textures forming layered structures are rare. Layered compound slopes are widely distributed in the northeastern hilly and gully areas of the Loess Plateau in China. These slopes are mostly composed of

sandstone (feldspathic and argillaceous sandstone) structures covered with thin layers of loess (Figure 1).

**Figure 1.** Geographical location of the study area and real images of sandstone covered with loess.

The change in the texture and interbedded structures of sandstone overlaid with loess modify the characteristics of water infiltration [12]. The following problems remain to be solved: the process of water infiltration in different types of sandstones overlaid with loess, and the effects of different interbedding patterns between feldspathic and argillaceous sandstones on water infiltration. Infiltration water prediction models mainly include the Van Genuchten (VG), Brooks–Corey (BC) [13], Philip, Kostiakov [14], and Horton [15] models. Hydrus-1D software is commonly used to establish models of soil water transport [16–18], and its simulation results have been proven to be reliable. However, the results of the predictive modeling of the infiltration water content of sandstone covered with loess based on the Hydru-1D model still need to be explored.

In accordance with the above problems, this study analyzed the soil water infiltration characteristics of four kinds of soil-covered sandstone through a one-dimensional vertical infiltration simulation experiment, as well as a comparative analysis of the changes in the four indicators of cumulative infiltration volume, infiltration rate, wetting front migration distance, and profile volume water content. On the basis of the Hydrus-1D model, this study retrieved the optimal hydraulic parameters and inverted the five parameters of the VG model to ensure accuracy. This study aims to provide a theoretical basis for the hydrological processes and sustainable utilization of water resources in the loess-overlaid sandstone slopes in the hilly and gully region of the Loess Plateau in China.

## 2. Materials and Methods

### 2.1. Experimental Design

Four types of soil-covered sandstones were identified in this experiment (Figure 2): loessial soil–feldspathic sandstone double-layered structure (L–FS), loessial soil–argillaceous sandstone double-layered structure (L–AS), loessial soil–feldspathic sandstone–argillaceous sandstone three-layered structure (L–FS–AS), and loessial soil–argillaceous sandstone–feldspathic sandstone structure (L–AS–FS). The loess and sandstone mentioned in this experiment were artificially manufactured with basic physical properties, such as bulk density and mechanical composition, in line with their natural state to facilitate the control of variables. The infiltration test device was a cylindrical container made of Plexiglass with the height of 80 cm and the diameter of 15 cm. Sieve plates were placed at the top 10 cm and the bottom of the container for water infiltration and free drainage. Three water probes were drilled at uniform distances (10, 30, and 50 cm) on the side of the container to monitor the volumetric water content dynamically. The other end of each probe was connected to a HoBo data collector to collect data. A Mariotte bottle was placed above the cylindrical container, and a hose was connected to supply water to the container.

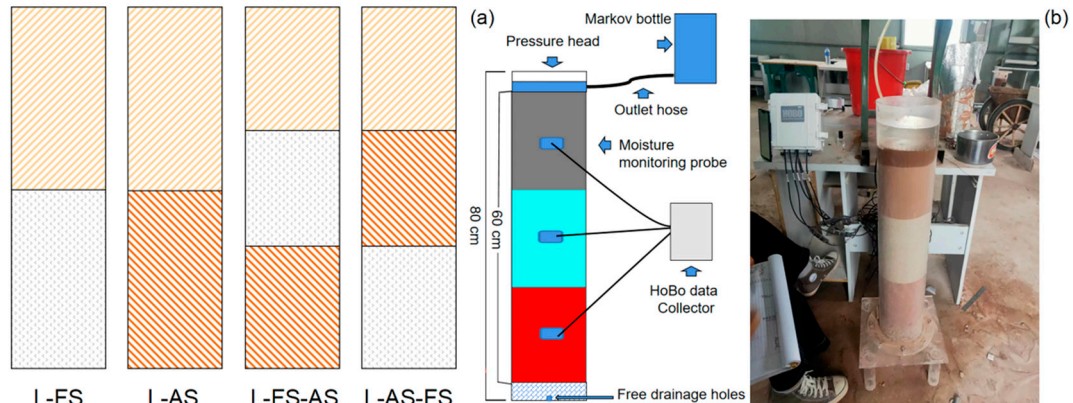

**Figure 2.** Schematic of the one-dimensional vertical infiltration test. (**a**) Schematic of the types of soil-covered sandstone required for the test. (**b**) Test device.

Feldspathic sandstone, loessial soil, and argillaceous sandstone debris were collected from the slope of the study area, crushed and air-dried indoors, then passed through a 2 mm soil sieve. At the time of sampling, a cutting ring and an aluminum box were used to take rock and soil samples at every 10 cm of soil depth. The samples were returned to the laboratory for the determination of the physical properties of each tested type of rock and soil (Table 1). The amount of soil filled in each layer was calculated in accordance with the physical index, and the calculation method is shown as Formula (1). In this study, on the basis of the common vertical stratification of loess and sandstone on the slope, the soil depth of 60 cm soil was selected to fill soil samples. Each type of container was filled with 5 cm of soil samples per layer, and the layers were roughened to make the soil samples in the one-dimensional vertical infiltration simulation test close to the natural state. Sand and gravel were laid at the bottom of the soil column to enhance the drainage effect. A pressure head of 3 cm was set in accordance with the depth of incoming water generated by rainfall on the unit area of slope surface in the study area. The final value of each test parameter was the average value of three measurements.

$$M = V \cdot B(1+W) \tag{1}$$

**Table 1.** Basic physical properties of the filled soil samples.

| Type | Bulk Density (g·cm$^{-3}$) | Mechanical Composition Content (%) | | | Initial Moisture Content (cm$^3$·cm$^{-3}$) | Total Porosity (%) |
|---|---|---|---|---|---|---|
| | | Clay <0.002 mm | Powder 0.05~0.002 mm | Sand 0.050~2.00 mm | | |
| L | 1.36 ± 0.09 | 23.62 ± 2.54 | 29.06 ± 1.14 | 65.02 ± 5.55 | 0.25 ± 0.04 | 40.48 ± 2.11 |
| FS | 1.68 ± 0.1 | 11.72 ± 1.36 | 5.88 ± 2.41 | 82.4 ± 7.12 | 0.27 ± 0.21 | 30.66 ± 3.45 |
| AS | 1.54 ± 0.13 | 27.21 ± 2.21 | 13.89 ± 5.01 | 58.9 ± 4.31 | 0.31 ± 0.07 | 37.4 ± 0.98 |

Note: L is loessial soil, FS is feldspathic sandstone and AS is argillaceous sandstone.

In the formula, M represents the mass of the filled soil (g), V represents the volume of the soil (cm$^3$), B represents the dry unit weight of soil mass (g/cm$^3$), and W represents the water content of soil (%).

### 2.2. Model Principle

This work, the law governing vertical water movement in loess–sandstone of different types was simulated on the basis of the prediction model of Hydrus-1D software. The main goal of modeling is the accurate reflection of water movement in soil under realistic conditions. Therefore, the key to the accuracy of a model lies in the accuracy of the hydraulic parameters of the relevant soil after calibration [19]. This requirement calls for paying close attention to the steps of parameter inversion. In this study, the soil water

movement parameters $\theta_x$, $\theta_s$, $\alpha$, $n$, and $K_s$ were adjusted and optimized numerous times during parameter inversion optimization, thus improving the match between the hydraulic parameters and the physical properties under natural conditions.

The hydraulic governing equation of the Hydras-1D one-dimensional water transport model is the improved Richards equation [19,20]:

$$\frac{\partial \theta}{\partial t} = \frac{\partial}{\partial z}\left[ K(\theta)\frac{\partial h}{\partial z} - K(\theta) \right] - S \tag{2}$$

In this formula, $\theta$ is the soil volumetric water content (cm$^3$/cm$^3$); $h$ is the matrix potential (cm); $t$ is the infiltration time (d); $K$ is the hydraulic conductivity (cm/d); $z$ is the vertical coordinate axis with the ground surface as the origin, and the direction vertically downward is positive (cm); and $S$ is the absorption and confluence term of plant roots (cm$^3$/cm$^{-3}\cdot$day$^{-1}$). In this study, the experimental soil designed was nonvegetated surface soil. Therefore, water absorption by roots did not exist. Thus, its value was set as 0.

The equation required five parameters of the soil water transport model: $\theta_x$, $\theta_s$, $\alpha$, $n$, and $K_s$. The key parameters of the model based on these five parameters were mainly derived from the soil water characteristic curve. Given that the soil in the study was unsaturated, the VG model was used to simulate the soil hydraulic parameters. The model is expressed as follows:

$$\theta(h) = \begin{cases} \theta_x + \dfrac{\theta_s - \theta_r}{[1+\alpha|h|^n]^m} & , h < 0 \\ \theta_s & , h \geq 0 \end{cases} \tag{3}$$

$$K(h) = K_s S_e^l \left[ 1 - \left( 1 - S_e^{\frac{1}{m}} \right)^m \right]^{-2} \tag{4}$$

$$S_e = (\theta - \theta_x)/(\theta_s - \theta_x)$$
$$m = 1 - \frac{1}{n}, n > 1 \tag{5}$$

In the formula: $\theta(h)$ is the water content corresponding to the pressure head (cm$^3$/cm$^3$); h is the soil pressure head (cm); $\theta x$ is the residual volumetric water content (cm$^3$/cm$^3$); $\theta s$ is the saturated volumetric water content (cm$^3$/cm$^3$); $\theta$ is the volumetric water content (cm$^3$/cm$^3$); $\alpha$, $n$, $m$ and $\iota$ are empirical constants, where $\alpha$ is the reciprocal of the intake air value (cm$^{-1}$), $n$ is the pore distribution coefficient, $\iota$ is the connectivity coefficient of soil pores, usually taking the empirical value of 0.5; $Ks$ is the saturated hydraulic conductivity of soil (m/d). $K(h)$ is the unsaturated hydraulic conductivity of soil (m/d); $Se$ is the effective water content of soil (cm$^3$/cm$^3$).

## 2.3. Accuracy of the Verification Parameters

The determination coefficient R$^2$ and relative error RE were used to verify the accuracy and rationality of the hydraulic parameters and initial conditions in the simulation [20,21]. The determination coefficient R$^2$ is used to reflect the deviation and coincidence degree of the curve between the measured and simulated values. Its value is generally between 0 and 1, and R$^2$ values close to 1 are indicative of the high degree of coincidence between simulated and measured values. The relative error RE can reflect the relative error between the total amount of measured and simulated values. Generally, if the RE is close to 0, then the fitting accuracy of the simulated and measured values is high. Its calculation formula is as follows:

$$R^2 = 1 - \frac{\sum_{i=1}^{n}[I(s)_i - I(o)_i]^2}{\sum_{i=1}^{n}[I(o)_i - I(o)]^2} \tag{6}$$

$$RE = \left| 1 - \frac{\sum_{i=1}^{n} I(s)_i}{\sum_{i=1}^{n} I(o)_i} \right| \tag{7}$$

In the formula: $I(o)_i$ represents the measured value, $I(s)_i$ represents the simulated value, and $I(o)$ represents the average value of $I(o)_i$ ($i = 1, 2, ..., n$).

## 3. Results

### 3.1. Characteristics of Water Infiltration in Loess–Sandstone Structures

Given that the infiltration rate can indicate the infiltration volume per unit time in the process of water infiltration, it is an important indicator for studying soil water infiltration [22,23]. With the increase in time, the infiltration rate showed an "L"-shaped change that was marked by rapid decline–slow decrease–stabilization (Figure 3). Table 2 shows that the change in the infiltration rate of various types of soil within 0–50 min of the initial infiltration rate followed the order of L–AS (0.91 cm/min) > L–FS (0.65 cm/min) > L–AS–FS (0.53 cm/min) > L–FS–AS (0.47 cm/min). The initial infiltration rate of L–AS was the fastest and was 1.4 times that of L–FS, 1.94 times that of L–FS–AS, and 1.72 times that of L–AS–FS. Moreover, the initial infiltration rate of the double-layered sandstone covered with loess was faster than that of the three-layered structured. The stable infiltration rate reflected the final state of the rate during the whole infiltration process. Figure 3 illustrates the trend of the stable infiltration rate during 150–450 min. It shows that the curve of L–AS fluctuated greatly and that the infiltration rate of other loess–sandstone types showed little fluctuation. Table 2 illustrates that the stable infiltration rate followed the order of L–AS (0.03 cm/min) > L–AS–FS (0.025 cm/min) > L–FS (0.015 cm/min) > L–FS–AS (0.01 cm/min). All of these results indicated that the transition layer between the loess and sandstone in the loess–sandstone structure caused a reduction in the water infiltration rate, thus affecting water infiltration ability. The cumulative infiltration volume will be analyzed and the above conclusion will be verified in the future.

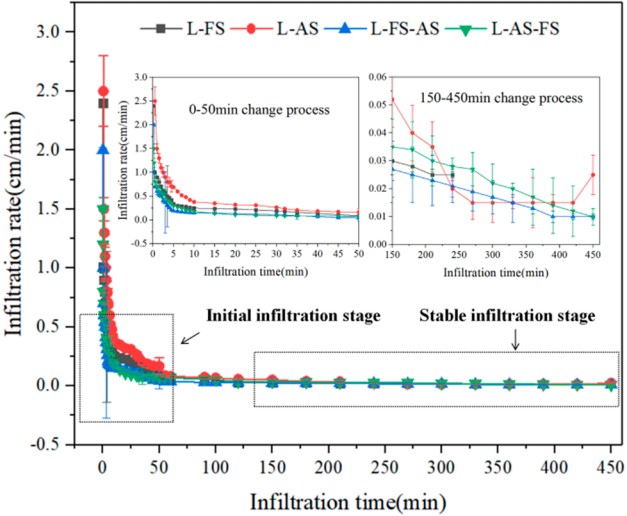

**Figure 3.** Variation process of vertical water infiltration rate.

**Table 2.** Characteristic parameters of vertical infiltration rate of water of different types of loess-sandstone structure.

| Type | Initial Infiltration Rate (cm·min$^{-1}$) | Stable Infiltration Rate (cm·min$^{-1}$) |
| :---: | :---: | :---: |
| L–FS | 0.65 | 0.025 |
| L–AS | 0.91 | 0.03 |
| L–FS–AS | 0.47 | 0.01 |
| L–AS–FS | 0.53 | 0.015 |

Note: The values in the table are the mean of the results of this experiment.

The cumulative infiltration volume is an important indicator for studying soil water infiltration because it can represent the overall infiltration volume up to a certain point of time or period in the process of soil water infiltration [24]. Figure 4 shows that with the prolongation of time, the cumulative infiltration volume of the loess–sandstone structures

increased, whereas the slope of the curve of cumulative infiltration volume decreased gradually. This phenomenon was related to the trend of the infiltration rate. As the infiltration rate stabilized gradually, the variation in its corresponding cumulative infiltration volume decreased accordingly until it stabilized [25]. The final cumulative infiltration volume followed the order of L–AS (23.775 cm) > L–AS–FS (22.7 cm) > L–FS (19.6 cm) > L–FS–AS (15.25 cm).

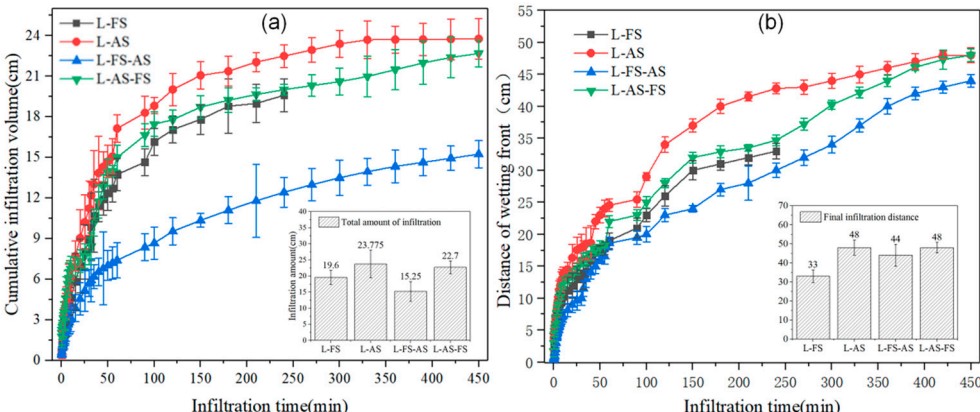

**Figure 4.** Vertical water infiltration volume and change of the wetting front. (**a**) is the change in cumulative infiltration volume, and (**b**) is the change in the wetting front.

The wetting front refers to the apparent interface that is formed between the moistened upper area of soil and the dry soil during water infiltration [26]. The downward movement of the wetting front results from the large gradient of water content in part of the soil of the wetting front. The matrix potential generated by this gradient can cause the continuous downward movement of the wetting front and determines the distance of water movement during soil water infiltration. Figure 4b shows that with time, the vertical migration distance of the wetting front increased, whereas the slope of the curve of the vertical migration distance of the wetting front gradually decreased. This phenomenon was related to the trend of infiltration rate. As the infiltration rate stabilized, the variation in the vertical migration distance of the corresponding wetting front diminished gradually until it increased cumulatively and steadily [27]. The final vertical movement distance of the wetting front followed the order of L–AS (48 cm) = L–AS–FS (48 cm) > L–FS–AS (33 cm) > L–FS (44 cm). Table 3 shows that the times for the wetting fronts of the L–FS and L–AS double-layered structures to reach the transition surface (30 cm) of loess and sandstone were 195 and 175 min, respectively. Moreover, the times needed for the wetting fronts of the L–FS–AS and L–AS–FS three-layered structures to reach the transition surface (20 cm) of loess and sandstone were 120 and 100 min, respectively, and to reach the transition surface (40 cm) between sandstone were 420 and 330 min, respectively. All of these findings indicated that the transition layer between loess and feldspathic in the loess–sandstone structure caused a reduction in the total amount of water infiltration and the downward speed of the wetting front, resulting in the decline in water infiltration ability.

**Table 3.** Time needed for wetting front reaching the interlayer transition surface.

| Type | | Location of Interlayer Transition Plane | Time (min) |
|---|---|---|---|
| Double layer structure | L–FS | 30 cm | 195 |
| | L–AS | 30 cm | 175 |
| Three layer structure | L–FS–AS | 20 cm | 120 |
| | | 40 cm | 420 |
| | L–AS–FS | 20 cm | 100 |
| | | 40 cm | 330 |

Note: The values in the table are the mean of the results of this experiment.

The above results clearly demonstrated that the water infiltration ability of L–FS and L–FS–AS structures was weak. Therefore, under the same upper inflow, water was prone to accumulate early on the surface of the layered structure at the same time or lateral seepage may occur in the cross-section of the soil layer. This phenomenon may be caused by the reduction in the permeability of soil with different textures. Some scholars posit that water decreases when it flows across soils of different textures. This effect is unrelated to the order of the overlying coarse-textured soil and fine-textured soil. The reduction in the permeability of coarse-textured soil overlaid with fine-textured soil is caused by hydraulic barriers. Meanwhile, the capillary barrier accounts for the phenomenon that occurs in fine-textured soil overlaid with coarse-textured soil. The transition surface of each layer of the L–FS and L–FS–AS loess–sandstone structures had two conditions: fine-textured soil overlaid with coarse-textured soil and coarse-textured soil overlaid with fine-textured soil. Therefore, water infiltration ability was affected successively by the capillary barrier and hydraulic barrier.

### 3.2. Simulation of Water Movement in Loess-Sandstone Structure

### 3.2.1. Determination and Inversion of Hydraulic Parameter of Soil

Given that the final significance of water infiltration characteristics of loess–sandstone is to reflect the dynamic change in water content, simulating the law governing water movement in loess–sandstone structures is of great importance for studying hydrological processes in the study area.

In this work, the mechanical composition data of the soil were obtained by using a Malvern 3000 laser particle analyzer on the basis of the soil samples of various textures obtained in situ in the field. After the percentage contents of clay, silt, and sand were measured in accordance with the American size grading standard (Table 1), the preliminary water movement parameter was extracted from the water flow model–soil hydraulic characteristic curve in the software preprocessing tool. At this point, the extracted parameters had not been calibrated. Therefore, the accuracy was low. The local minimum values of the hydraulic parameters of different structures are presented in Table 4. The hydraulic parameters of each type of loess–sandstone structure after adjustments and calibrations are given in Table 5.

**Table 4.** Local minimum values of hydraulic parameters for different structures.

| Type | Depth (cm) | $\theta_x$ (cm$^3$·cm$^{-3}$) | $\theta_s$ (cm$^3$·cm$^{-3}$) | $\alpha$ (cm$^{-1}$) | $n$ | $K_s$ (cm/s) |
|---|---|---|---|---|---|---|
| | 0–20 | 0.0362 | 0.2629 | 0.02528 | 1.329 | 0.00513 |
| L | 20–40 | 0.0311 | 0.24 | 0.04528 | 1.213 | $8.15 \times 10^{-4}$ |
| | 40–60 | 0.0332 | 0.256 | 0.04525 | 1.229 | $1.15 \times 10^{-4}$ |
| | 0–20 | 0.031 | 0.2626 | 0.0332 | 1.6508 | 0.00753 |
| FS | 20–40 | 0.028 | 0.2724 | 0.02938 | 1.6508 | 0.00301 |
| | 40–60 | 0.031 | 0.1995 | 0.0332 | 1.4508 | $1.38 \times 10^{-4}$ |
| | 0–20 | 0.068 | 0.3412 | 0.01166 | 1.243 | $2.07 \times 10^{-4}$ |
| AS | 20–40 | 0.038 | 0.2286 | 0.01416 | 1.257 | $2.35 \times 10^{-4}$ |
| | 40–60 | 0.033 | 0.3122 | 0.02136 | 1.145 | $1.56 \times 10^{-4}$ |

### 3.2.2. Establishment of Spatial and Temporal Information and Setting Boundary Conditions

The spatial information setting was consistent with the one-dimensional vertical infiltration test. The model was set to simulate the soil depth of 60 cm, and every 0.5 cm was a vertical spatial step, that is, the soil layer was divided into 121 nodes in the vertical direction. Moreover, four loess-overlaid sandstone probabilistic models were set (Figure 5a) in accordance with the stratification characteristics of the loess–sandstone structures and the vertical distribution characteristics of soil dry density. The locations of the moisture monitoring points were laid out exactly in accordance the actual situation of the one-dimensional vertical infiltration test at the soil depths of 10, 30, and 50 cm. The monitoring

point was set to determine the initial moisture content of the loess and sandstone samples at the depth of 0–60 cm through the analysis of ANOVA results (Figure 5b). The time settings were based on the consistent duration of the one-dimensional vertical infiltration test on each soil type, and the time step was chosen to be every 1 s. The number of time nodes was set to N+1, and N was the total time duration. The minimum number of iterations was set to 0.01, and the maximum number of iterations was set to 10. The boundary conditions were "constant pressure heads" for the upper boundary matching the one-dimensional vertical infiltration test, and the head height was set to 3 cm. Given that the depth of the water table in the study area is unclear but that its geographical location is known to be in the gully area of Chinese loess hills and forests, the lower boundary condition was selected as "free drainage" in reference to the current situation that the depth of the water table in Chinese loess hills is 40–100 m. In addition, given that soil volumetric water content was monitored indoors, evaporation was not considered.

**Table 5.** Hydraulic parameters of soil bodies of different structures optimized by inversion.

| Type | Depth (cm) | $\theta_x$ (cm$^3 \cdot$cm$^{-3}$) | $\theta_s$ (cm$^3 \cdot$cm$^{-3}$) | $\alpha$ (cm$^{-1}$) | $n$ | $K_s$ (cm/s) |
|---|---|---|---|---|---|---|
| | 0–20 | 0.04795 | 0.3271 | 0.06071 | 1.438 | 0.00803 |
| L–FS | 20–40 | 0.038 | 0.2808 | 0.05166 | 1.543 | 0.0576 |
| | 40–60 | 0.028 | 0.2799 | 0.05166 | 1.543 | 0.0221 |
| | 0–20 | 0.05795 | 0.2833 | 0.03528 | 1.696 | 0.00639 |
| L–AS | 20–40 | 0.038 | 0.3562 | 0.0423 | 1.427 | $6.92 \times 10^{-4}$ |
| | 40–60 | 0.035 | 0.4786 | 0.01899 | 6.408 | $4.49 \times 10^{-4}$ |
| | 0–20 | 0.05795 | 0.265 | 0.0922 | 1.929 | 0.00518 |
| L–FS–AS | 20–40 | 0.045 | 0.2724 | 0.02938 | 1.75 | 0.00301 |
| | 40–60 | 0.033 | 0.332 | 0.05155 | 1.584 | 0.005 |
| | 0–20 | 0.05795 | 0.2824 | 0.03149 | 1.849 | 0.00538 |
| L–AS–FS | 20–40 | 0.038 | 0.2306 | 0.02863 | 4.31 | $2.91 \times 10^{-4}$ |
| | 40–60 | 0.035 | 0.2802 | 0.03541 | 1.589 | $1.96 \times 10^{-4}$ |

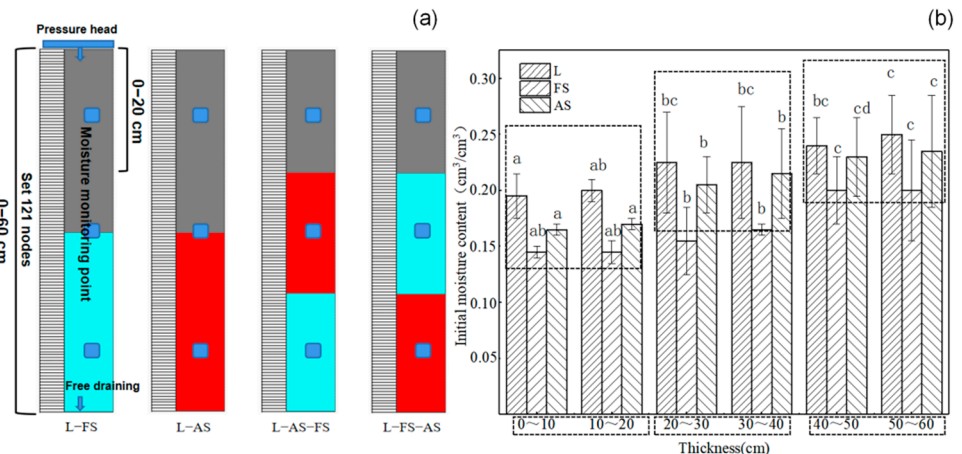

**Figure 5.** Model discrete schematic diagram and monitoring point setting basis. (**a**) is the schematic of profile simulation and boundary settings. (**b**) is the initial moisture content of the loess and sandstone samples at the depths of 0–60 cm determined by analyzing ANOVA results.

### 3.2.3. Simulation and Accuracy Verification of Moisture Transport in Loess-Overlaid Sandstone

On the basis of the above optimized inverse performance of the parameters of the soil moisture transport model for each type of loess-overlaid sandstone structure in different soil layers (Table 5), the assigned values of $\theta_x$, $\theta_s$, $\alpha$, $n$, and $K_s$ for each modeling scenario were introduced into the established model to acquire the simulation curves of soil moisture transport in each soil configuration then compared with the measured profile volume water content curves obtained in the one-dimensional vertical test (Figure 6).

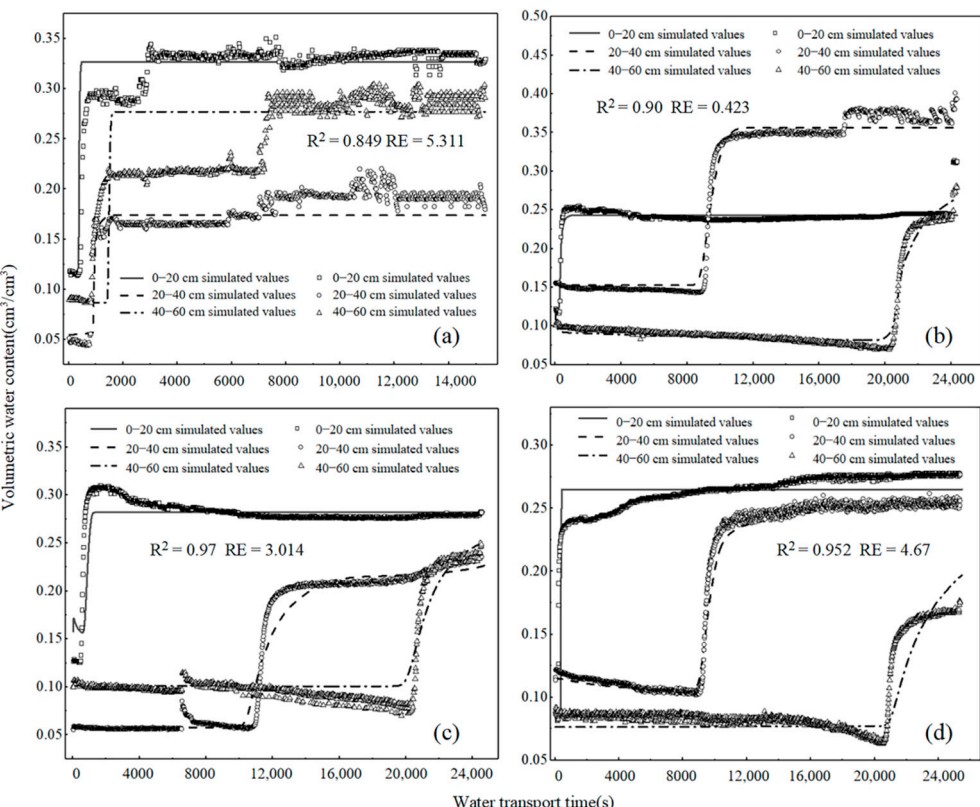

**Figure 6.** Simulation of water vertical transport in different types of loess-overlaid sandstones. (**a**) is the change process of the L–FS structure, (**b**) is the change process of the L–AS structure, (**c**) is the change process of the L–FS–AF structure, and (**d**) is the change process of the L–AS–FS structure.

Overall, the variation pattern of each simulated curve for the 0–20 cm soil layer was L-shaped, and the variation trend of the 20–60 cm soil layer was S-shaped. The overall change curve of the soil volumetric water content in the 0–20 cm soil layer during the whole water transport process demonstrated a brief rapid increase then stabilized. In the L–AS and L–AS–FS structures, the volumetric water contents of the 20–40 cm soil layer changed from being lower than those of the 0–20 cm soil layer for a period of time in the early stage to surpassing those of the 0–20 cm soil layer at a certain time before stabilizing. In the L–FS and L–FS–AS structures, the moisture content of the 20–40 cm soil layer was lower than that of the 40–60 cm layer for most of the time during the first period. The coefficient of determination $R^2$ between the measured and simulated values of soil moisture for each type of loess-overlaid sandstone ranged from 0.849 to 0.97. The relative error RE values were below 5.311% and were distributed between 0.423–5.311%. These results showed that the measured values of each type of loess-overlaid sandstone fit well with the simulated values. The validation accuracy of L–FS was lower than that of other loess-overlaid sandstones, indicating that the inversion parameters and the accuracy control of the test process in the model of this type of loess-overlaid sandstone need to be improved.

## 4. Discussion

### 4.1. Influencing Factors of Water Infiltration in Loess-Overlaid Sandstone

This work studied infiltration in loess-overlaid sandstone with different textural compositions and found that the trends of infiltration rate, accumulated infiltration volume, and moisture front distance curves were consistent with those reported by other works [28–31]. Some scholars have also investigated the influencing factors of soil moisture infiltration, such as the effects of changes in capacitance, initial water content, pressure head, soil depth, and void size on moisture infiltration. The increase in soil bulk density leads to a decrease in cumulative infiltration, infiltration rate, and wetting frontage distance [32].

The cumulative infiltration and average infiltration rates of soil decrease gradually with the increase in soil moisture content, exhibiting a negative correlation [33]. Some scholars believe that the cumulative infiltration of water is positively correlated with the pressure head [34]. Several studies have demonstrated that the increase in porosity increases the ability of water to penetrate [35,36]. The transport of water in soil voids is affected by the combined effect of the pressure head, initial water content, bulk density, and other factors, and the weight of each factor affects the law governing water transport. In the future, the influencing factors of loess-overlaid sandstone should be studied thoroughly from these perspectives. In addition, the reduction in infiltration found in this work is only a theoretical description and lacks quantitative data support. For example, the pattern of soil hydraulic parameters in the interlayer variation and the mechanism underlying the effect of capillary porosity on interlayer water infiltration require further study. Finally, in the natural state, the vertical infiltration of water may also produce lateral migration due to the influence of layered structures. During the simulation, water infiltration occasionally produced side flow along the inner wall of the Plexiglass container. Both of the above conditions can cause water loss, which can affect the accuracy of the experimental results. In the future, efforts should be made to address these issues.

### 4.2. Role of Infiltration in Loess-Overlaid Sandstone in Hydraulic Erosion

The slope surface consists of the overlying soil and the underlying bedrock [37–39], and the slope surface is the main site wherein hydraulic erosion occurs [40–42]. Hydraulic erosion runoff is generated by the rainfall–vegetation retention–surface infiltration–evaporation–storage of full (hyperinfiltration) flow production [43–45]. Therefore, slopes with widely distributed overburdening sandstone may also have more complex hydraulic erosion processes than homogeneous slopes. Therefore, the questions of how much the water infiltration process in loess-overlaid sandstone contributes to the overall hydraulic erosion process and which type of loess-overlaid sandstone is capable of producing flow should also be systematically studied.

### 5. Conclusions

In this work, the water vertical infiltration characteristics and water content variation patterns of different types of loess-overlaid sandstones were simulated. Given that the interlayer transition surfaces of loess–feldspathic sandstone and loess–feldspathic sandstone–argillaceous sandstone coincided with fine soil overlaid with coarse soil and coarse soil overlaid with fine soil, water infiltration decreased, and water transport capacity was successively affected by capillary barriers and hydraulic barriers. The existence of a transition layer of loess and feldspathic sandstone in the loess-overlaid sandstone structures reduced the water infiltration rate, total water infiltration amount, and downward movement rate of the wetting front. These effects further affected the water infiltration capacity. On the basis of the Hydrus-1D model, 15 sets of soil hydraulic parameters, namely, $\theta_x$ (0.028–0.05795 cm$^3$/cm$^3$), $\theta_s$ (0.2306–0.4786 cm$^3$/cm$^3$), $\alpha$ (0.01899–0.06071 cm$^{-1}$), $n$ (1.438–6.408), and $Ks$ (1.96·10$^{-4}$–0.0576 cm/s), were inverted and optimized for each 20 cm soil layer (total of 60 cm). The VG model constructed by applying these parameters exhibited high accuracy in simulating the vertical infiltration of moisture content in the loess-overlaid sandstone structures with the coefficient of determination R$^2$ > 0.849 and the relative error RE < 5.311. This work is important for the study of the hydrological and soil erosion processes in loess-overlaid sandstone slopes.

**Author Contributions:** F.Q. and X.D. designed this study; X.D. conducted the field work, performed the data analysis, and wrote the first draft of the manuscript; L.L., Z.Y., Y.L. and Y.W. improved the English language and grammatical editing. All authors have read and agreed to the published version of the manuscript.

**Funding:** Funding was provided by the project of Major project of Natural Science Foundation of Inner Mongolia Autonomous Region(2021ZD07), the Universities Young Scientific and Technological

Talents of the Inner Mongolia autonomous region (NJYT22046), the central government to guide local scientific and technological development (No. 2021ZY0023), the National Natural Science Foundation of China (grant No. 42267049), the Fundamental Research Funds for China Institute of Water Resource and Hydropower Research (MK2021J06).

**Data Availability Statement:** Not applicable.

**Conflicts of Interest:** The authors declare no conflict of interest.

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
