# Peer review of "Study on Water Vertical Infiltration Characteristics and Water Content Simulation of Sandstone Overlying Loess"

_water, doi:10.3390/w14223716_

Round 1
Reviewer 1 Report
Manuscript number:water-2000457
Title of manuscript:Study on water vertical infiltration characteristics and water content simulation of sandstone overlying loess
Comments and suggestions for modification:
In this paper, one-dimensional vertical experiment and Hydrus-1D model are used to simulate the water vertical infiltration law and water vertical transport of the thin layer loess-sandstone structure covered by the slope in the loess hilly and gully region of China, and the five optimal hydraulic parameters in the VG model are retrieved.The scientific problem to be solved in this paper is relatively clear, the structure of this paper is complete, and the research method has certain reference value, but there are still shortcomings in this paper:
(1) The ABSTRACT of the article is lack of quantitative information, please add it.
(2) It is mentioned in the fourth paragraph of the INTRODUCTION that the prediction model of moisture content is established in this paper, but this part does not appear in the main text. Please check the manuscript carefully.
(3) Carefully check the use of "space" in the article, some content in the article does not use space after punctuation;
(4) Symbols representing parameters in the paper should be in italics (modified in the full text of the abstract, formula, Table, etc.), and standard errors are not indicated in Table 2 and Table 3, or "mean" is reflected in the title of the table;
(5) In Figure 6 (d), "value" or "values" should be unified;
(6) It is suggested that the author should draw according to the author's submission guide of Water.
(7) There is a mistake in the subtitle mark of the article. It is suggested to check and correct the serial number of the title at all levels of the article.
(8) Only three monitoring points are set for soil moisture measurement in the article. Why is this choice? Is there any limitation for soil parameter inversion?
(9) The reasons why VG model is chosen should be explained in 2.2;
(10) Part of the discussion should be expanded. For example, in "4.1", the influencing factors of vertical water infiltration should be discussed in detail, such as the influence of a certain influencing factor on the infiltration characteristics and its laws; Finally, the influence of water loss during the experiment on the overall results of the study should be reflected in the discussion.

Author Response
Dear reviewer:
Thank you for your decision and constructive comments on my manuscript. We have carefully considered the suggestion of reviewer and make some changes. We have tried our best to improve and made some changes in the manuscript. According to your comments. Revision notes, point-to-point, are given as follows:
Point 1: The ABSTRACT of the article is lack of quantitative information, please add it.
Response 1: Lines 25-30 of the Word version have been modified. The optimal hydraulic parameters for each type in the range of 0-60 cm soil layers have been added to the abstract ” Based on the Hydrus-1D model, a total of 15 sets of soil hydraulic parameters θx (0.028-0.05795 cm3/cm3), θs (0.2306-0.4786 cm3/cm3), α (0.01899-0.06071 cm-1), n (1.438-6.408) and Ks (1.96·10-4-0.0576 cm/s) were inverted and optimized for each 20 cm soil layer (total 60 cm)”.
Point 2: It is mentioned in the fourth paragraph of the INTRODUCTION that the prediction model of moisture content is established in this paper, but this part does not appear in the main text. Please check the manuscript carefully.
Response 2: Replaced the original "established a prediction model of infiltration moisture content to assure accuracy." with "inverted the 5 parameters of the VG model to ensure accuracy. " This article mentions that "model establishment" means that the VG model is used as a template, and five important parameters in the model are obtained through inversion optimization.
Point 3: Carefully check the use of "space" in the article, some content in the article does not use space after punctuation;
Response 3: This type of error has been modified in the article.
Point 4: Symbols representing parameters in the paper should be in italics (modified in the full text of the abstract, formula, Table, etc.), and standard errors are not indicated in Table 2 and Table 3, or "mean" is reflected in the title of the table;
Response 4: The error related to "italic" in the text has been fixed; The description of "Note: The values in the table are the mean of the results of this experiment." has been added below the tables 2 and 3.
Point 5: In Figure 6 (d), "value" or "values" should be unified;
Response 5: The relevant content in the figure has been unified as "values".
Point 6: It is suggested that the author should draw according to the author's submission guide of Water.
Response 6: In accordance with the author's guidelines, the format of the picture has been modified and the resolution has been increased.
Point 7: There is a mistake in the subtitle mark of the article. It is suggested to check and correct the serial number of the title at all levels of the article.
Response 7: The serial number of the title has been checked and corrected for various levels of the article.
Point 8: Only three monitoring points are set for soil moisture measurement in the article. Why is this choice? Is there any limitation for soil parameter inversion?
Response 8: The three monitoring points selected in this document are not set arbitrarily. Our team took samples from the study area at a depth of 60 cm and measured the initial moisture content of the unsifted soil and sandstone by drying method (results of 3 repeated trials). The results showed no significant difference in saturated moisture content per 20cm depth (Figure 5-b). In addition, the sandstone of various types of loess is analyzed as the overall structure of the water migration simulation experiment. Therefore, we set up three monitoring points at 10cm, 30cm, and 50cm with a deep level of 20cm. In summary, a new image of "Initial moisture content of the loess and sandstone samples in the depth of 0-60cm by analysis of the results of ANOVA" has been added to the article.
For “Is there any limitation for soil parameter inversion?” My answer to the question is as follows:
There is indeed a minimum value limit for each parameter, which is not reflected in my article. Moreover, the data in Table 4 reflect the simulated values of loess and sandstones obtained using HYDRUS software, not the actual minimum values. The values in Table 4 are only used to compare with the final inversion optimization results (Table 5) and have no other significance. Therefore, we have changed the contents of Table 4 to the actual measured local minimum.
|
Table.4 Local minimum values of hydraulic parameters for different structures. |
||||||
|
Type |
Depth(cm) |
θx(cm3·cm-3) |
θs(cm3·cm-3) |
α(cm-1) |
n |
Ks(cm/s) |
|
L |
0-20 |
0.0362 |
0.2629 |
0.02528 |
1.329 |
0.00513 |
|
20-40 |
0.0311 |
0.24 |
0.04528 |
1.213 |
8.15E-04 |
|
|
40-60 |
0.0332 |
0.256 |
0.04525 |
1.229 |
1.15E-04 |
|
|
FS |
0-20 |
0.031 |
0.2626 |
0.0332 |
1.6508 |
0.00753 |
|
20-40 |
0.028 |
0.2724 |
0.02938 |
1.6508 |
0.00301 |
|
|
40-60 |
0.031 |
0.1995 |
0.0332 |
1.4508 |
1.38E-04 |
|
|
AS |
0-20 |
0.068 |
0.3412 |
0.01166 |
1.243 |
2.07E-04 |
|
20-40 |
0.038 |
0.2286 |
0.01416 |
1.257 |
2.35E-04 |
|
|
40-60 |
0.033 |
0.3122 |
0.02136 |
1.145 |
1.56E-04 |
|
Note: when the decimal digits are more than 6, the scientific notation should be adopted.
Then I found that some of the parameters in Table 5 did not end at the local minimum(Same as you said), so I re-inverted the results. The results of the final modifications are shown in the table below. Correspondingly, this results in a decrease in the accuracy of the final moisture simulation (Figure 6), and the results are still good( R2>0.849 and RE<5.311).
|
Table.5 Hydraulic parameters of soil bodies of different structures optimized by inversion. |
||||||
|
Type |
Depth(cm) |
θx(cm3·cm-3) |
θs(cm3·cm-3) |
α(cm-1) |
n |
Ks(cm/s) |
|
L-FS |
0-20 |
0.04795 |
0.3271 |
0.06071 |
1.438 |
0.00803 |
|
20-40 |
0.038 |
0.2808 |
0.05166 |
1.543 |
0.0576 |
|
|
40-60 |
0.028 |
0.2799 |
0.05166 |
1.543 |
0.0221 |
|
|
L-AS |
0-20 |
0.05795 |
0.2833 |
0.03528 |
1.696 |
0.00639 |
|
20-40 |
0.038 |
0.3562 |
0.0423 |
1.427 |
6.92E-04 |
|
|
40-60 |
0.035 |
0.4786 |
0.01899 |
6.408 |
4.49E-04 |
|
|
L-FS-AS |
0-20 |
0.05795 |
0.265 |
0.0922 |
1.929 |
0.00518 |
|
20-40 |
0.045 |
0.2724 |
0.02938 |
1.75 |
0.00301 |
|
|
40-60 |
0.033 |
0.332 |
0.05155 |
1.584 |
0.005 |
|
|
L-AS-FS |
0-20 |
0.05795 |
0.2824 |
0.03149 |
1.849 |
0.00538 |
|
20-40 |
0.038 |
0.2306 |
0.02863 |
4.31 |
2.91E-04 |
|
|
40-60 |
0.035 |
0.2802 |
0.03541 |
1.589 |
1.96E-04 |
|
Point 9: The reasons why VG model is chosen should be explained in 2.2;
Response 9: The soil structure of the slope in the study area is a water unsaturated structure. The VG model was chosen because it can fit the soil moisture characteristic curve θ(h) and water conductivity K(h) in the unsaturated state, which is widely used.
Point 10: Part of the discussion should be expanded. For example, in "4.1", the influencing factors of vertical water infiltration should be discussed in detail, such as the influence of a certain influencing factor on the infiltration characteristics and its laws; Finally, the influence of water loss during the experiment on the overall results of the study should be reflected in the discussion.
Response 10: The revised Word version of lines 343-346 and 351-353 of this submission is the place where the changes have been made. Following your suggestion, the relevant content has been added to section 4.1 of the article.

Author Response
Dear reviewer:
Thank you for your decision and constructive comments on my manuscript. We have carefully considered the suggestion of reviewer and make some changes. We have tried our best to improve and made some changes in the manuscript. According to your comments. Revision notes, point-to-point, are given as follows:
Point 1: The use of English language is very poor. I started correcting it, but soon found it would take to much time, so I’d recommend to have the paper corrected by a person specialized in scientifific English papers!
Response 1: We apologize for the poor language of our manuscript. We worked on the manuscript for a long time and the repeated addition and removal of sentences and sections obviously led to poor readability. We have now worked on both language and readability and have also involved native English speakers for language corrections. We really hope that the flow and language level have been substantially improved.
Point 2: The manuscript has a section ”introduction” and then immediately continues with ”results”. I think the readability would be improved when authors follow the classic sections: introduction, materials and methods, results, discussion!
Response 2: The manuscript has been changed in the form "introduction, materials and methods, results, discussion".
Point 3: It is very confusing. Author sometimes speak about optimizing the parameters of the top layer, at other times they optimize the parameters of all layers.
Response 3: This manuscript optimizes the parameters of all layers (0-60cm), as mentioned in the article "The description of optimizing the parameters of the top layer" is wrong and I have corrected it(All the relevant statements have been changed throughout the text, and the values of the three layers of soil for each type of soil have also been changed to Table 4).
Point 4: When authors are optimizing 3 layers, it means they have 15 parameters to optimize. Chances are they end in a local minimum!
Response 4: There is indeed a minimum value limit for each parameter, which is not reflected in my article. Moreover, the data in Table 4 reflect the simulated values of loess and sandstones obtained using HYDRUS software, not the actual minimum values. The values in Table 4 are only used to compare with the final inversion optimization results (Table 5) and have no other significance. Therefore, we have changed the contents of Table 4 to the actual measured local minimum.
|
Table.4 Local minimum values of hydraulic parameters for different structures. |
||||||
|
Type |
Depth(cm) |
θx(cm3·cm-3) |
θs(cm3·cm-3) |
α(cm-1) |
n |
Ks(cm/s) |
|
L |
0-20 |
0.0362 |
0.2629 |
0.02528 |
1.329 |
0.00513 |
|
20-40 |
0.0311 |
0.24 |
0.04528 |
1.213 |
8.15E-04 |
|
|
40-60 |
0.0332 |
0.256 |
0.04525 |
1.229 |
1.15E-04 |
|
|
FS |
0-20 |
0.031 |
0.2626 |
0.0332 |
1.6508 |
0.00753 |
|
20-40 |
0.028 |
0.2724 |
0.02938 |
1.6508 |
0.00301 |
|
|
40-60 |
0.031 |
0.1995 |
0.0332 |
1.4508 |
1.38E-04 |
|
|
AS |
0-20 |
0.068 |
0.3412 |
0.01166 |
1.243 |
2.07E-04 |
|
20-40 |
0.038 |
0.2286 |
0.01416 |
1.257 |
2.35E-04 |
|
|
40-60 |
0.033 |
0.3122 |
0.02136 |
1.145 |
1.56E-04 |
|
Note: when the decimal digit are more than 6, the scientific notation should be adopted.
Then we found that some of the parameters in Table 5 did not end at the local minimum(Same as you said), so we re-inverted the results. The results of the final modifications are shown in the table below. Correspondingly, this results in a decrease in the accuracy of the final moisture simulation (Figure 6), and the results are still good( R2>0.849 and RE<5.311).
|
Table.5 Hydraulic parameters of soil bodies of different structures optimized by inversion. |
||||||
|
Type |
Depth(cm) |
θx(cm3·cm-3) |
θs(cm3·cm-3) |
α(cm-1) |
n |
Ks(cm/s) |
|
L-FS |
0-20 |
0.04795 |
0.3271 |
0.06071 |
1.438 |
0.00803 |
|
20-40 |
0.038 |
0.2808 |
0.05166 |
1.543 |
0.0576 |
|
|
40-60 |
0.028 |
0.2799 |
0.05166 |
1.543 |
0.0221 |
|
|
L-AS |
0-20 |
0.05795 |
0.2833 |
0.03528 |
1.696 |
0.00639 |
|
20-40 |
0.038 |
0.3562 |
0.0423 |
1.427 |
6.92E-04 |
|
|
40-60 |
0.035 |
0.4786 |
0.01899 |
6.408 |
4.49E-04 |
|
|
L-FS-AS |
0-20 |
0.05795 |
0.265 |
0.0922 |
1.929 |
0.00518 |
|
20-40 |
0.045 |
0.2724 |
0.02938 |
1.75 |
0.00301 |
|
|
40-60 |
0.033 |
0.332 |
0.05155 |
1.584 |
0.005 |
|
|
L-AS-FS |
0-20 |
0.05795 |
0.2824 |
0.03149 |
1.849 |
0.00538 |
|
20-40 |
0.038 |
0.2306 |
0.02863 |
4.31 |
2.91E-04 |
|
|
40-60 |
0.035 |
0.2802 |
0.03541 |
1.589 |
1.96E-04 |
|
Point 5: Authors pick up the debris of sandstone, crush it, sieve it and then compact it to a certain density. And keep calling it sandstone. In my opinion they should clearly mention that the materials in the soil column are artifificial.
Response 5: The explanation of "The loess and sandstone mentioned in this experiment were artificially manufactured with basic physical properties, such as bulk density and mechanical composition, in line with their natural state to facilitate the control of variables." has been added to the second sentence of the first paragraph of article 2.1 (Line 80-82 of Word version).
Point 6: Table 5 presents the results of the optimization procedures. However, when I look at the parameters corresponding to the same soil type, I can see quite some difffferent values (as expected!). But what does this mean for the results when talking about the parameters of sandstone?
Response 6: You can see from Table 5 that the same soil type has different parameters, which is common in water migration simulations such as Hydrus-1d or RETC. The main reasons are as follows:
- Different soil depths will lead to differences in the values of hydraulic parameters of the same structure type.
- In the simulation of soil water migration law, many models are often used, including Van Genuchten(VG) model, Brooks-Corey(BC) mode, Philips model, Horton model, etc., And there are steps to determine and calculate hydraulic parameters in the application of the above models. Each hydraulic parameter presents a different value, so that the final model calculation result is close to the actual value. That is to say, the ultimate purpose of each parameter inversion process is not to reflect the parameters themselves (Of course, the inversion process of parameters is limited by the local local minimum and maximum values.), but to reflect the comprehensive influence of all parameters on the calculation results of the model. Therefore, there will be cases where the parameters of the same type of soil will be different.
So far, this has been the case with many published academic works, as follows:
① Yang, Y., Chen, Y., Chen, J., Zhang, Z., Li, Y., & Du, Y. (2021). The applicability of HYDRUS‐1D to infiltration of water‐repellent soil at different depths. European Journal of Soil Science, 72(5), 2020–2032.
② lqbal, M., Kamal, M. R., M., M. F., Che Man, H., & Wayayok, A. (2020). HYDRUS-1D Simulation of Soil Water Dynamics for Sweet Corn under Tropical Rainfed Condition. Applied Sciences, 10(4), 1219.
③ Caiqiong, Y., & Jun, F. (2016). Application of HYDRUS-1D model to provide antecedent soil water contents for analysis of runoff and soil erosion from a slope on the Loess Plateau. CATENA, 139, 1–8.
In summary, the parameter values obtained by the inversion of Table 5 in this study are not intended to illustrate the parameter distribution of sandstone in the study area, but as empirical parameters for predicting the water transport law of loess sandstone structures based on VG models. The results show that the VG model uses the values of the parameters in Table 5 as the empirical parameters to simulate the results, and the calculated vertical migration law of water can well indicate the actual situation.
Point 7: In equation 3 authors use some unusual characters around the h. I guess they mean the absolute value of h, indicated by |h|.
Response 7: The error in Equation 3 has been fixed. (Line 137 of Word version)
Point 8: In line 130 authors state that θ(h) is the change of water content with soil water pressure. This is not true! θ(h) is the water content corresponding to the pressure head h!
Response 8: The description of "θ(h) is the change of soil volumetric water content with soil water potential (cm3/cm3)" in line 142 of Word version has been revised to "θ(h) is the water content corresponding to the pressure head (cm3/cm3)".
Point 9: Line 236-243 should be moved to either introduction or materials and methods.
Response 9: Lines 236-243 have been moved to the section on materials and methods(Lines 116-123 of Word version).

Round 2
Reviewer 2 Report
Authors did respond correctly to my remarks and questions, improving the manuscript.